# The magnitude of chronic diabetes complications and its associated factors among diabetic patients attending the general hospitals in Tigray region, Northern Ethiopia

**Kalayou K. Berhe** [1]*, **Lilian T. Mselle**[2], **Haftu B. Gebru**[1]

1 School of Nursing, College of Health Sciences, Mekelle University, Mekelle City, Tigray Region, Northern Ethiopia, 2 Department of Clinical Nursing, School of Nursing, Muhimbli University of Health and Allied Sciences, (MUHAS), Dar es Salaam, Tanzania

* Kalushaibex@gmail.com

## Abstract

### Background

Diabetes is a severe challenge to global public health since it is a leading cause of morbidity, mortality, and rising healthcare costs. 3.0 million Ethiopians, or 4.7% of the population, had diabetes in 2021. Studies on the chronic complications of diabetes in Ethiopia have not been conducted in lower-level healthcare facilities, so the findings from tertiary hospitals do not accurately reflect the issues with chronic diabetes in general hospitals. In addition, there is a lack of information and little research on the complications of chronic diabetes in Ethiopia. The objective of this study was to assess the degree of chronic diabetes complications and associated factors among diabetic patients presenting to general hospitals in the Tigray area in northern Ethiopia.

### Methods

As part of a multi-centre cross-sectional study, 1,158 type 2 diabetes (T2D) patients from 10 general hospitals in the Tigray region were randomly chosen. An interviewer-administered questionnaire, a record review, and an SPSS version 20 analysis were used to collect the data. All continuous data were presented as mean standard deviation (SD), while categorical data were identified by frequencies. Using a multivariable logistic regression model, the factors associated with chronic diabetes complications among T2D diabetic patients were found, and linked factors were declared at p 0.05.

### Results

Fifty-four of people with diabetes have chronic problems. Hypertension (27%) eye illness, renal disease (19.1%), and hypertension (27%) eye disease were the most common long-term effects of diabetes. Patients with chronic diabetes complications were more likely to be older than 60, taking insulin and an OHGA (Oral Hyperglycemic Agent) (AOR = 3.00; 95% CI 1.73, 5.26), having diabetes for more than five years, taking more than four tablets per

**Funding:** The author(s) received no specific funding for this work.

**Competing interests:** The authors have declared that no competing interests exist.

**Abbreviations:** AOR, Adjusted Odd ratio; COR, Crud Odd Ratio; CI, Confidence Interval; BMI, Body Ma Index; EFY, Ethiopian Fiscal Year; FB, Fasting Blood Sugar; HDL, High-Density Lipoprotein; LDL, Low-Density Lipoprotein; AMP, Amputation; OHA, Oral Hypoglycaemic Agent; RB, Random Blood Glucose; SD, Standard Deviation; SPSS, Statistical Package for Social Science; T2D, Type 2 Diabetes; TIA, Transit Ischemic Attach; WHO, World Health Organization; BP, Blood Pressure; BP, Systolic Blood Pressure; DBP, Dystonic Blood Pressure; HgbA1C, Glycolated Hgb; IRB, Institutional Review Board; ERC, Ethical Review Committee; IDF, International Diabetes Federation and mmHg: mm of mercury.

day (AOR = 1.63; 95% CI 1.23,2.15), and having high systolic and diastolic blood pressure. Patients with government employment (AOR = 0.48; 95% CI 0.26, 0.90), antiplatelet drug use (AOR = 0.29; 95% CI 0.16, 0.52), and medication for treating dyslipidemia (AOR = 0.54; 95% CI 0.35, 0.84), all had a decreased chance of developing a chronic diabetes problem.

## Conclusion

At least one chronic diabetic complication was present in more than half of the patients in this study. Chronic diabetes problems were related to patients' characteristics like age, occupation, diabetes treatment plan, anti-platelet, anti-dyslipidemia medicine, duration of diabetes, high Systolic BP, high Diastolic BP, and pill burden. To avoid complications from occurring, diabetes care professionals and stakeholders must collaborate to establish appropriate methods, especially for individuals who are more likely to experience diabetic complications.

## Background

Hyperglycemia is a metabolic condition associated with diabetes mellitus [1]. There are two basic forms of diabetes: Type 1 diabetes and Type 2 diabetes. Type 1 diabetes mainly affects youngsters and is defined by an insulin shortage that necessitates daily insulin injections [2]. The most prevalent form of diabetes with insulin resistance and a relative insulin deficit is type 2 [3]. There are now more cases of diabetes than ever before due to population growth, age, urbanization, obesity, and inactivity [4]. The prevalence of diabetes was found to be 10.5% (537 million people) worldwide, 4.5% (24 million people) in Africa, and 4.7% (3 million people) in Ethiopia, according to an IDF (International Diabetes Federation) atlas report from 2021 [5].

Diabetes impacts people's functional skills and quality of life, leading to severe morbidity and premature mortality, and is one of the top public health concerns in the globe [6]. It negatively affects socioeconomic development and public health globally [7]. [8] Diabetes accounts for more than 80% of all early fatalities from non-communicable diseases (NCDs), along with cardiovascular disease, cancer, and respiratory illnesses [8]. In addition, patients with diabetes have a two to three times higher risk of dying from any cause, including cancer, cardiovascular disease, stroke, chronic renal disease, and liver disease [9–11]. Due to insufficient monitoring and skewed or incorrect laboratory data, patients are more likely to have limited compliance with therapies in outpatient settings [12]. Diabetic individuals with poorly controlled hyperglycemia can develop macrovascular diseases such as peripheral artery disease, cardiovascular disease, and cerebrovascular disease as well as microvascular consequences like retinopathy, nephropathy, and neuropathy [13,14].

For instance, studies on the prevalence of chronic diabetic complications revealed that 96% of patients had hypertension, 46% had peripheral neuropathy, 30% had neuropathy, and 7% had neuropathy encountered impotence [15] and the main reason for admission was diabetic foot ulcer (39%) and cardiovascular disease (21%) [16]. Diabetic retinopathy is a prominent cause of blindness, of which 2.6% is attributed to diabetes [17]. In addition, glaucoma, cataract, and other disorder of the eye occur earlier and more frequently in people with diabetes [18]. Similarly, there is a remarkable prevalence of both acute and chronic complications in diabetic cases in Ethiopia [19].

Information on the prevalence of diabetes-related complications is essential to change diabetes management policies and practices to effectively control the disease. However, studies focusing on such topics are rare in Ethiopia [15,20,21] and no studies have been conducted in the study area. In Ethiopia, diabetes-related studies mainly focused on glycemic control, dyslipidemia, self-care management, diabetes prevalence and diabetes complications in hospitalized/high-risk patients [22–26].

Although very few studies on chronic diabetes complications have been conducted in different parts of the country, there has not been a recent comprehensive study of outpatients in general hospitals. On the other hand, because those studies were conducted in tertiary hospitals, they were unable to provide a precise picture of diabetes complications at a lower level, such as in general hospitals. In addition, those studies were carried out among high-risk individuals, the study population of type 1 and 2 diabetes, assessing only acute complications, microvascular or macrovascular complications, and in some studies data were collected through document review only. Furthermore, it is essential to investigate diabetes complications and their contributing factors regularly to pot evolving patterns and formulate diabetes management strategies. Therefore, this study aimed to identify chronic complications related to diabetes and associated factors in general hospitals in the Tigray region of Northern Ethiopia.

## Method and materials

### Study design, setting and period

A multi-centre cross-sectional study was conducted in the Tigray region from September 2019 to January 2020. Tigray is one of the ten regional states of Ethiopia. The Ethiopian health care system is organized into three-tier: primary, secondary and tertiary levels of care. The primary level of care is provided at a primary hospital, health centre and health post. The Primary Health care Unit (PHCU) is composed of a health centre (HC) and five satellite health posts (HP). These facilities provide service to approximately 25,000 people. A primary hospital provides inpatient and ambulatory service to an average of 100,000 Population and has an inpatient capacity of 25–50 beds. A general hospital provides inpatient and ambulatory services to an average of 1,000,000 people. A tertiary hospital serves an average of five million people. It serves as a referral to the general hospital [27].

In 2019/2020, in the Tigray region, there were 2 referral hospitals, 14 general hospitals, 24 primary hospitals, 230 health centres and 741 health posts. There were more than 310 ambulances and a well-established referral system. There were over 750 private health facilities in the private sector, ranging from drug vendors and clinics to general and specialized hospitals. There were more than 25,000 health workforce ranging from health extension workers to specialists and sub-specialists [28]. This study was done in ten selected public general hospitals namely Alamata, Lemlem Carl, Mekelle, Adigrat, kids Mariam, Adwa, Abiyi Adi, Shule Shire, Sheraro Mayani and Kahsay Abera (Humera). These general hospitals provide basic health services for patients with different diseases including diabetes mellitus. About 4,154 patients with type 2 diabetes received health services at these public general hospitals in 2018/19 [29].

**Population.**   All type 2 diabetic (T2D) patients admitted to the study hospitals and those who attended diabetic clinics during the data collection period participated in the study. To be involved in the study participants had to be adult.

### Eligibility

**Inclusion criteria.**   All adult patients aged more than 18 years who were diagnosed to have T2D and had follow-up visits in the study hospitals for ≥1 year. The study did not include patients.

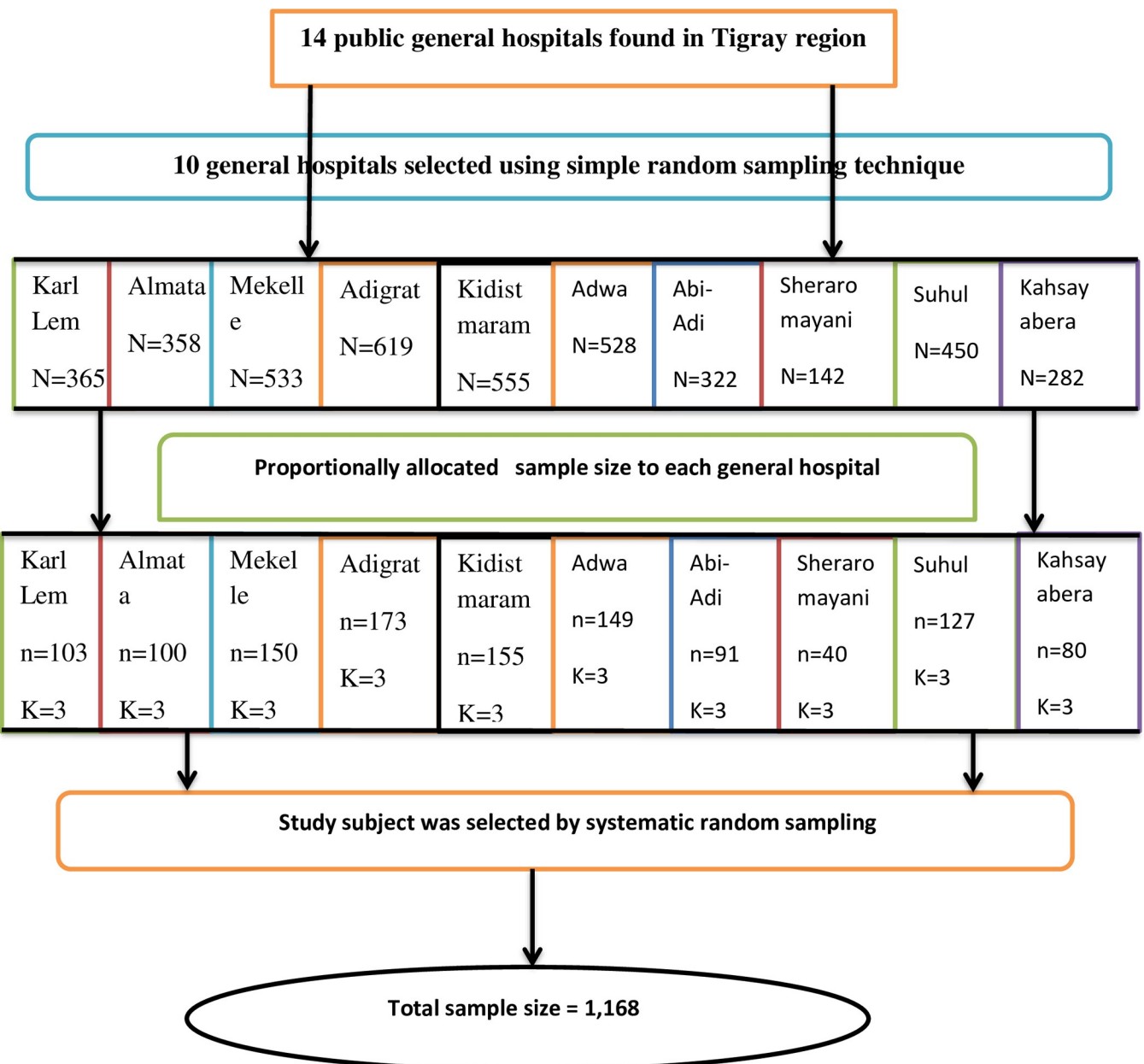

**Fig 1. Schematic presentation of the sampling procedure for a study on magnitude of chronic diabetes complication among diabetic patients attending the general hospitals in Tigray region, Northern Ethiopia, 2019/2020.**

**Exclusion criteria.** All Patients who were pregnant and critically ill.

**Sample Size determination.** The sample size (n) was estimated using a single population proportion formula proposed by Cochran [30] with an assumption of 95% confidence interval (z = 1.96), a margin of error of 0.03 and the proportion of chronic diabetes complication among T2D patients from a study in Northwestern Ethiopia which was 53.5% (P = 0.535) [21]. The initial sample size was 1,061 which was obtained using the following formula ni = (Z1-α/2)2p (1–p)/d2 = (1.96)$^2$0.535(1–0.535)/(0.03)$^2$. 10% of the initial sample size was added for non-response rate and the final sample size was 1,168. Proportional allocation of participants

was employed to allocate the sample size among the selected general hospitals based on case-load (Fig 1).

**Sampling procedure.** Ten out of fourteen public general hospitals were selected through a simple random sampling technique, and all general hospitals were not included because of budget constraints. Participants were selected using a systematic random sampling method whereby the first patient was selected randomly from the first three by a lottery method, and the next patient was selected every three intervals until the required sample was attained.

**Data collection tool and measurement.** The data were collected using a pre-tested, interviewer-administered questionnaire that was developed based on relevant literature [12,21,31,32] and record review. It has four parts: part one was used to collect data about socio-demographic characteristics; part two was on clinical characteristics; part three was on behavioural factors; and part four was on chronic diabetes complications. Behavioural variables were assessed based on the WHO TEPwise approach for chronic disease risk factor surveillance [33]. Clinical variables were taken from the patient's chart/medical record and physical measurements. Body weight was measured to an accuracy of 0.1 kg using a weight scale machine and the patient was barefoot and wearing light clothing. Height was measured in meters, standing upright on a flat surface, by a stadiometer. Body Mass Index (BMI) was calculated as the ratio of weight in kilogram (kg) to the square of height in meters (m2). Systolic blood pressure (BP) and diastolic blood pressure (DBP) was measured from the left arm at the level of the heart using a mercury-based or digital sphygmomanometer after the patient took a rest for more than 10 minute and 1–2 hour for those who took hot drink like coffee [34]. For those patients with BP $\geq$ 140 mm of mercury (mmHg) and/or DBP $\geq$ 90 mmHg, blood pressure was measured again, and finally, the average value was taken.

**Data collection procedure.** The T2D patients attending 10 hospitals during the data collection period were approached by the data collectors, verified for eligibility and then, after informed consent was obtained, data were collected. Data was collected by ten BSc nurses who had either MPH or MSc with multilingual abilities and were supervised by the first and third authors. The type 2 diabetic patients were identified by examining their diagnosis as reported in the medical record. Moreover, clinical and chronic diabetic complications data were extracted from patients' medical records. The diagnosis of chronic diabetic complication was then confirmed by the physician.

## Data management and analysis

To assure data quality, training and orientation of the study were done for the data collectors and supervisors and the questionnaire was pre-tested and checked for its validity and reliability. The pretest of the questionnaire involved 2% of the sample size and it was carried out in Quiha general hospital two weeks before the actual data collection. The questionnaire was revised based on the pre-test results. The questionnaires were checked for completeness and consistency on the daily bases. The data was entered and cleaned, entered in the SPSS version 20 analyses were performed using. A binary logistic regression analysis model was used to identify factors associated with chronic diabetes complications. The Homer-Lemehow goodness-of-fit test was used to check the model fitness and the assumption of a P-value >0.05 was considered a good model fit. Independent variables with p < 0.20 during the bivariate analysis were then included in the multivariable logistic regression for further analysis to control confounding factors. Multicollinearity between independent variables was checked by using the tolerance test and variance inflation factor (VIF). P < 0.05 was considered the cut-off point for reporting an independent variable that shows a statistically significant association with the dependent variable in multivariate analysis. The strength of the association of factors with

chronic diabetes complications was demonstrated by computing the adjusted odd ratio (AOR) and its 95% confidence interval (CI).

## Variables of the study

### Independent variable

Socio-demographic: age, sex, marital status, religion, ethnicity, educational status, occupation, residence, monthly income (UD), family history of diabetes and BMI.

Clinical: diabetes treatment regimen, anti-platelet drug (e.g., AA), anti-dyslipidemia drug, glucometer, duration of diabetes, Fasting Blood Glucose (FBG), Systolic Blood Pressure (SBP), Diastolic Blood Pressure (DBP), Pill burden, and high health care cost.

Behavioural: adherence to a diabetic diet, saturated fat consumption, vegetable consumption, adherence to diabetic medication, self-blood glucose test, smoking, alcohol consumption, physical activity, and diabetes education.

**Dependent variable.** Chronic diabetes complications status

**Operational definition.** The data about chronic diabetes complications were extracted from the patient's chart/medical record. Only chronic complications that developed after the diagnosis of T2D and could be attributed to diabetes were considered in this study.

**Diagnosis of chronic diabetes complication.** Hypertension: was defined as systolic blood pressure (SBP) $\geq$ 140 mmHg and/or diastolic blood pressure (DBP) $\geq$ 90 mmHg, and/or patient on antihypertensive therapy was taken as the hypertensive patient [35,36].

Coronary artery disease (CAD): The diagnosis criteria for CAD were either a patient with typical anginal pain or equivalent symptoms or an abnormal resting ECG or an asymptomatic patient with the abnormal stress test, either by ECG or echo or a nuclear perfusion imaging test [37].

Peripheral vascular disease: The presence of intermittent claudication and/or an ABI (ankle-brachial/Arm index) value (< 0.9) in any limb was recorded as peripheral vascular disease [38].

Neuropathy: was diagnosed if a change was found in two or more of the three items hypesthesia or anaesthesia in lower and upper limbs when the patient's lower limp was evaluated [18].

Eye diseases: Eye diseases such as cataracts, glaucoma and diabetic retinopathy were identified based on the report of the ophthalmologist or optometrist from the dilated eye (fundus photography for retinopathy), and comprehensive eye examination, which was recorded on the patient's chart [18].

Chronic Kidney disease: was diagnosed based on the presence of urinary albumin(micro or macro albuminuria), and/or an abnormally high level of serum creatinine (low glomerular filtration rate) [39].

Foot problem: The diagnosis of foot problem was made through foot examination for any abnormalities (i.e. dry kin, fissure, deformities, callosities, ulceration, prominent vein, and nail lesion) or all patients were asked about a history of foot ulcer, neuro ischemic foot, or amputation [40].

Follow a healthy diet: consuming vegetables, beans and peas, fruit, whole grain, nut, and seeds, seafood, low-fat milk and milk product, and a moderate amount of lean meat, poultry, and egg [41,42].

High fat/oil consumption: eat or consume more than 10% of calories from saturated fat, which means more than 20 grams of saturated fat per day [43,44].

Alcohol consumption: Adult with diabetes who drinks alcohol should do so in moderation (no more than one drink per day for adult women and no more than two drinks per day for

adult men). One drink is defined as 12 oz/355ml of beer, 5 oz /148ml glass of wine, or 1.5 oz/ 44ml of distilled spirit [45].

Physical activity: At least 150 minutes per week of aerobic exercise, plus at least two seasons per week of resistance exercise, are recommended [46].

**Ethical considerations.** Ethical approval to conduct the study was obtained from the Institutional Review Board (IRB) of Mekelle University (Ref No. ERC 1370/2019). The study received approval from the Tigray regional health bureau and permission from the Medical Directors of the 10 involved hospitals. The study was conducted following the declaration of Helsinki. Study participants were recruited voluntarily after they were informed about the study and that they can withdraw from the study at any stage thereafter they signed the consent form. All data were kept in a safe and secure place anonymously to ensure confidentiality and only the researchers had access to the data.

## Results

### Socio-demographic characteristics of the participant

Overall, a total of 1,168 diabetic individuals were eligible for the study, but only 1,158 of the participant's questionnaires were fit for final analysis, which makes the response rate 99.14 per cent. Thirty-four per cent of the participants had an age greater than 60 years, with a mean age of 55.9 (D± 11.9) years. Most of the participants were male (54%), Married (67%), Orthodox Christian (88%), Tigrian ethnicity (96.3%), and urban resident (72.3%). Majority of the participants: 50.5% had no formal education, 80.3% had no family history of diabetes, 29.7% were unemployed, 7.9% had a monthly income of $34.26-$171.16 and 76.5% had a BMI of <25 kg/m2 [Table 1].

### Clinical and behavioural characteristics of the participants

Of the total patients included in this study, 78.9%, 11.7%, and 16.8% were taking oral hypoglycemic agent (OHGA), anti-coagulant drug, and anti-dyslipidemia drug respectively. The mean duration of diabetes was 6.3 (D ± 4.6) years and 54.6% of participants had diabetes duration of < 5 years. Of all participants, 10.1% had a glucometer at home, 60.7% had FBG of >130.00 mg/dl, 25.8% had SBP of >149.00 mmHg and 9.0% had DBP of > 90.00 mmHg. From the total participants, 42.7% were taking >4 pills per day, 86.4% had a high health care cost, 69.7% consumed a high amount of saturated fat (>20 g per day), and 68.7% were taking less than the recommended amount of vegetable (<4 serving per week). Of all participants, 90.6% and 50.4% adhered to diabetes medication and a healthy diet. Of the total participants, nine out of ten test their blood glucose once per month, 6.2% ever smoke tobacco product, 12.4% consume more than moderate amount of alcohol (≥3 drink per day). Moreover, of the total participants, 61.3% were active physically, and 76.5% were attending diabetes education at the time of follow visit [Table 2].

### The magnitude of chronic diabetes complication

Overall, 54% of participants (95% CI 51.35, 57.10) suffered from at least one chronic complication of diabetes. Among all participants who had the complication, 10.5% had a single complication, 16.9% lived with two and 26.8% with more than two types of complications respectively.

**Macrovascular complication.** Of all types of macrovascular complications, hypertension 27% (95% CI 24.71,29.85) was the most common but cerebrovascular disease 4.31% (95% CI:

**Table 1. Socio-demographic characteristics by chronic diabetes complication among diabetic patients attending the general hospitals in Tigray region, Northern Ethiopian, 2019/2020 (N = 1,158).**

| Variable | Category | Chronic DM complication status | | Total |
|---|---|---|---|---|
| | | No (n = 530) | Yes (n = 628) | |
| Age | ≤40 year | 88(7.6%) | 49(4.2%) | 137(11.8%) |
| | 41–45 year | 59(5.1%) | 41(3.5%) | 100(8.6%) |
| | 46–50 year | 96(8.3%) | 89(7.7%) | 185(16.0%) |
| | 51–55 year | 76(6.6%) | 76(6.6%) | 152(13.1%) |
| | 56–60 year | 93(8.0%) | 103(8.9%) | 196(16.9%) |
| | ≥61 year | 118(10.2%) | 270(23.3%) | 388(33.5%) |
| Sex | 1. Male | 285(24.6%) | 340(29.4%) | 625(54.0%) |
| | 2. Female | 245(21.2%) | 288(24.9%) | 533(46.0%) |
| Marital status | 1. ingle | 39(3.4%) | 38(3.3%) | 77(6.6%) |
| | 2. Married | 380(32.8%) | 394(34.0%) | 774(66.8%) |
| | 3. Divorced | 47(4.1%) | 84(7.3%) | 131(11.3%) |
| | 4. Widowed | 64(5.5%) | 112(9.7%) | 176(15.2%) |
| Religion | 1. Orthodox | 474(40.9%) | 545(47.1%) | 1019(88.0%) |
| | 2. Muslim | 50(4.3%) | 79(6.8%) | 129(11.1%) |
| | 3. Catholic | 6(0.5%) | 4(0.3%) | 10(0.9%) |
| Ethnicity | 1. Tigrian | 508(43.9%) | 607(52.4%) | 1115(96.3%) |
| | 2. Amhara | 19(1.6%) | 19(1.6%) | 38(3.3%) |
| | 3. Afar | 3(0.3%) | 2(0.2%) | 5(0.4%) |
| Educational status | 1. No formal education | 246(21.2%) | 339(29.3%) | 585(50.5%) |
| | 2. Primary school (1–8 grade) | 118(10.2%) | 134(11.6%) | 252(21.8%) |
| | 3. Secondary (9–12 grade) | 89(7.7%) | 80(6.9%) | 169(14.6%) |
| | 4. College/University | 77(6.6%) | 75(6.5%) | 152(13.1%) |
| Occupation | 1. Farmer | 119(10.3%) | 129(11.1%) | 248(21.4%) |
| | 2. Gov't employee | 103(8.9%) | 62(5.4%) | 165(14.2%) |
| | 3. Private work | 135(11.7%) | 146(12.6%) | 281(24.3%) |
| | 4. Retired | 42(3.6%) | 78(6.7%) | 120(10.4%) |
| | 5. Unemployed | 131(11.3%) | 213(18.4%) | 344(29.7%) |
| Residence | 1. Urban | 370(32.0%) | 467(40.3%) | 837(72.3%) |
| | 2. Rural | 160(13.8%) | 161(13.9%) | 321(27.7%) |
| Monthly income (UD) | < $34.23 | 157(13.6%) | 234(20.2%) | 391(33.8%) |
| | 2, $34.26–171.16 | 324(28.0%) | 351(30.3%) | 675(58.3%) |
| | 3, >$171.16 | 49(4.2%) | 43(3.7%) | 92(7.9%) |
| Family history of DM | 1.Ye | 102(8.8%) | 126(10.9%) | 228(19.7%) |
| | 2. No | 428(37.0%) | 502(43.4%) | 930(80.3%) |
| BMI | < 25 kg/m2 | 466(40.2%) | 420(36.3%) | 886(76.5%) |
| | ≥ 25 kg/m2 | 162(14.0%) | 110(9.5%) | 272(23.5%) |

T2D: Type 2 diabetes, DM: Diabetes mellitus, BMI: Body Ma Index, DM: Diabetes Mellitus, UD: United States Dollar.

3.14, 5.49) was the least common type. Out of all participants who had cerebrovascular complications, 0.3% had TIA and 4% had a history of stroke (Fig 2).

**Microvascular complication.** The most common type of microvascular complication was ocular disease 22.62% (95% CI 20.21, 25.03), followed by kidney disease 19.17% (95% CI 16.90, 21.44) and peripheral neuropathy11% (95% CI 8.84, 12.39) respectively. Patients with renal complications consisted of 9% with microalbuminuria,4.0% with macroalbuminuria and 6% with high levels of serum creatinine. Ocular complications included cataracts, retinopathy,

**Table 2. Clinical and behavioural characteristics by chronic diabetes complication among diabetic patients attending the general hospitals in Tigray region, Northern Ethiopian, 2019/2020 (N = 1,158).**

| Variable | Category | Chronic DM complication status | | Total |
|---|---|---|---|---|
| | | No (n = 530) | Yes (n = 628) | |
| Diabetes treatment regimen | Inulin (injectable) | 78(6.7%) | 61(5.3%) | 139(12.0%) |
| | Inulin & OHGA* | 33(2.8%) | 72(6.2%) | 105(9.1%) |
| | OHGA* | 419(36.2%) | 495(42.7%) | 914(78.9%) |
| Use of anti-platelet drug (e.g. AA) | 1, Ye | 20(1.7%) | 116(10.0%) | 136(11.7%) |
| | No | 510(44.0%) | 512(44.2%) | 1022(88.3%) |
| Use of anti-dyslipidemia drug | Ye | 45(3.9%) | 150(13.0%) | 195(16.8%) |
| | No | 485(41.9%) | 478(41.3% | 963(83.2%) |
| Have glucometer at home | Ye | 47(4.1%) | 70(6.0%) | 117(10.1%) |
| | No | 483(41.7%) | 558(48.2%) | 1041(89.9%) |
| Duration of diabetes since it occurred | < 5 year | 345 | 287(24.8%) | 632(54.6%) |
| | ≥ 5 year | 185(16.0%) | 341(29.4%) | 526(45.4%) |
| FBG test | <130.00 mg/dl | 200(17.3%) | 255(22.0%) | 455(39.3%) |
| | ≥ 130.99 mg/dl) | 330(28.5%) | 373(32.2%) | 703(60.7%) |
| Systolic blood pressure (SBP) | ≤ 139.99 mmHg | 459(39.6%) | 400(34.5%) | 859(74.2%) |
| | ≥140.00 mmHg | 71(6.1%) | 228(19.7%) | 299(25.8%) |
| Diastolic blood pressure (DBP) | ≤ 89.99 mmHg | 503(43.4%) | 551(47.6%) | 1054(91.0%) |
| | ≥90.00 mmHg | 27(2.3%) | 77(6.6%) | 104(9.0%) |
| Pill burden | <4 pills/day | 360(31.1%) | 304(26.3%) | 664(57.3%) |
| | ≥ 4 pills /day | 170(14.7%) | 324(28.0%) | 494(42.7%) |
| High healthcare cot | 1. Ye | 451(38.9%) | 550(47.5%) | 1001(86.4%) |
| | 2. No | 79(6.8%) | 78(6.7%) | 157(13.6%) |
| Adherence to a diabetic diet | Adhere | 264(22.8%) | 320(27.6%) | 264(22.8%) |
| | Not adhere | 266(23.0%) | 308(26.6%) | 266(23.0%) |
| Saturated fat consumption | < 20 gm. fat/day | 143(12.3%) | 208(18.0%) | 351(30.3%) |
| | ≥ 20 gm. fat/day | 387(33.4%) | 420(36.3%) | 807(69.7%) |
| Vegetable consumption per week | <4 serving | 370(32.0%) | 425(36.7%) | 795(68.7%) |
| | ≥ 4 serving | 160(13.8%) | 203(17.5%) | 363(31.3%) |
| Adherence to diabetic Medication | Adhere | 486(42.0%) | 563(48.6%) | 1049(90.6%) |
| | Not adhere | 44(3.8%) | 65(5.6%) | 109(9.4%) |
| Day, in which glucose was measured/wk. | Not measured at all | 488(42.1%) | 566(48.9%) | 1054(91.0%) |
| | 1–2 day | 42(3.6%) | 62(5.4%) | 104(9.0%) |
| Ever smoked tobacco products (smoking) | Ye | 27(2.3%) | 45(3.9%) | 72(6.2%) |
| | No | 503(43.4%) | 583(50.3%) | 1086(93.8%) |
| Alcohol consumption | ≥ 3 drinks per day | 62(5.4%) | 82(7.1%) | 144(12.4%) |
| | ≤ 2 drinks per day | 468(40.4%) | 546(47.2%) | 1014(87.6%) |
| Physical activity | Inactive | 174(15.0%) | 274(23.7%) | 448(38.7%) |
| | Active | 356(30.7%) | 354(30.6%) | 710(61.3%) |
| Attend diabetes education | Ye | 345(36.1%) | 387(40.4%) | 732(76.5%) |
| | No | 102(10.7%) | 123(12.9%) | 225(23.5%) |

*OHGA = Oral Hypoglycemic Agent, T2D: Type 2 diabetes, ASA: Acetylsalicylic Acid, FBG: Fasting Blood Glucose, SBP: systolic Blood Pressure, DBP: Diastolic Blood Pressure.

diabetes blindness and glaucoma, with magnitudes of 5%, 9%, 1% and 8% respectively. Of the total participant who had a foot disease, 4% had a diabetes-related foot ulcer, 1% had a foot amputation, 3.6% had ischemic pain, 1% had gangrene and 3% had an infection (Fig 3).

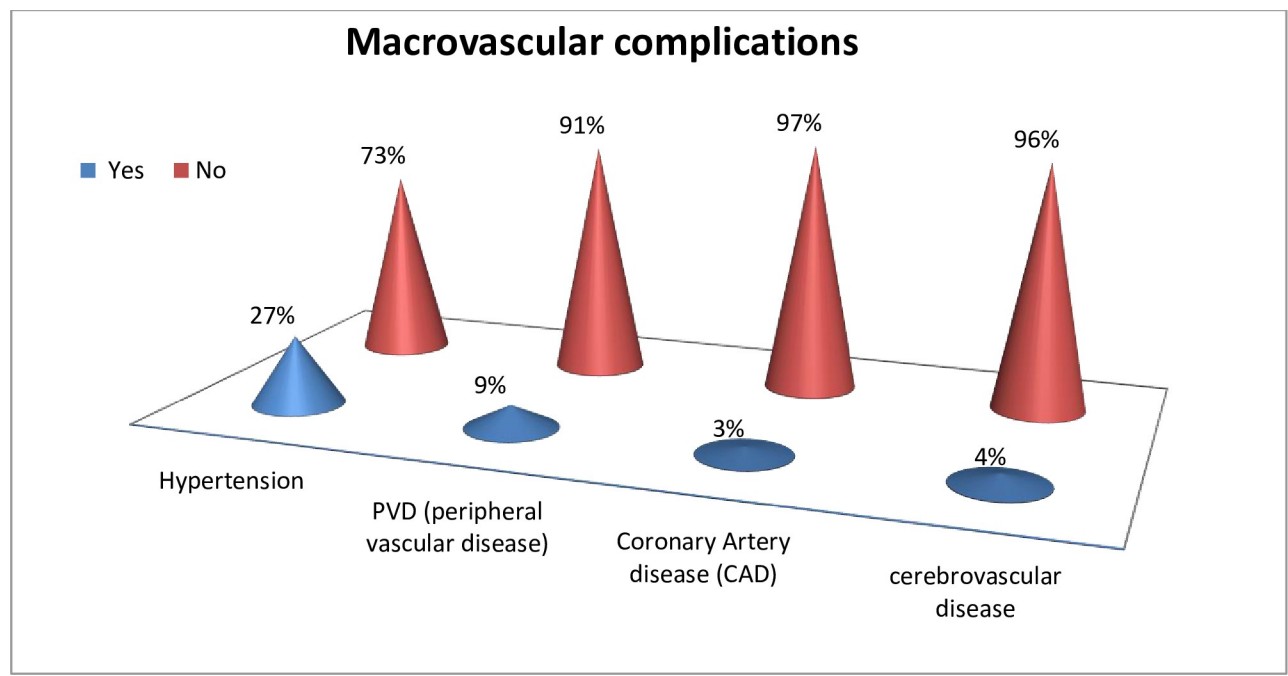

**Fig 2. Magnitude of macrovascular complications among diabetic patients attending the general hospitals in Tigray region, Northern Ethiopian, 2019/2020.**

## Factor associated with chronic diabetes complication among patients with type 2 diabetes (T2D)

The Homer-Lemehow goodness-of-fit test was done, and its result showed P = 0.0063, which was considered a good model fit. The odds ratio was calculated for factors found to be associated with chronic diabetes complications among type 2 diabetic patients. After considering all

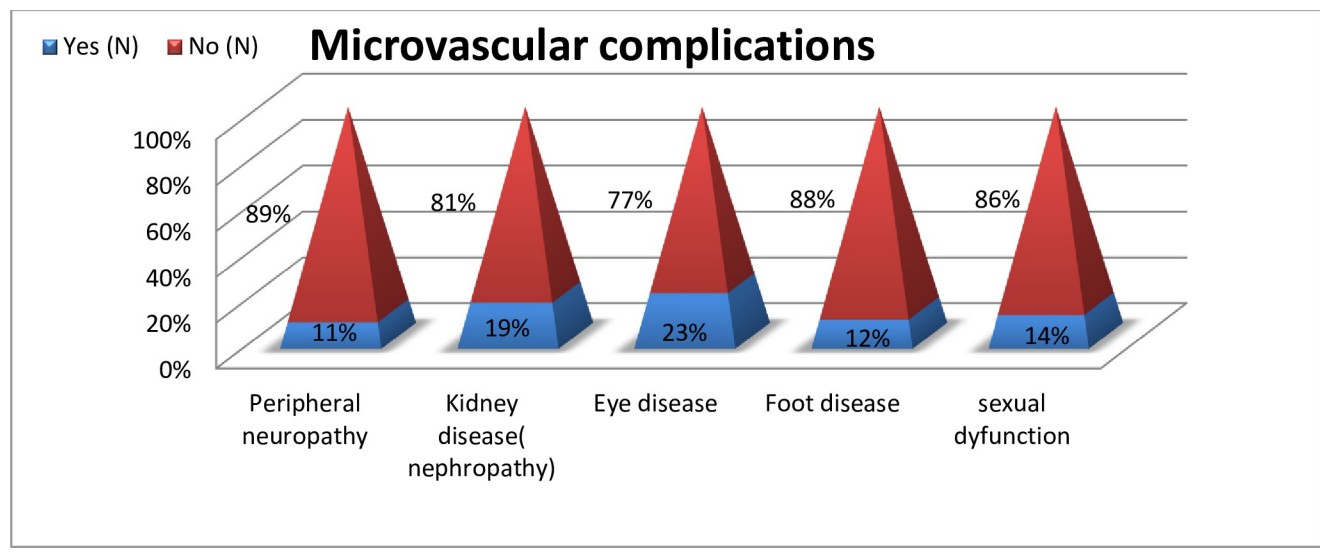

**Fig 3. Magnitude of microvascular complications among diabetic patients attending the general hospitals in Tigray region, Northern Ethiopian, 2019/2020.**

assumptions of binary logistic regression and the p-value ($\leq 0.05$) in the bivariate analysis, fifteen variables were identified as candidates for analysis in the multivariable model. In the multivariable logistic regression analysis, nine variables were found to be factors associated with chronic diabetes complications at a 5% level of significance.

The odds of a chronic diabetes complication was higher among patients age > 60 years (AOR = 3.00; 95% CI 1.73,5.26)) than their counterpart, who took insulin and OHA had a higher chance of developing the complication (AOR = 2.20; 95% CI 1.18, 4.27) than patients taking insulin injection only, with diabetes duration of $\geq 5$ years, were at higher risk to develop the complication (AOR = 1.56; 95% CI 1.18, 2.05) than patients with shorter diabetes duration and who took $\geq 4$ pills a day had a greater risk to develop chronic diabetes complication (AOR = 1.63; 95% CI 1.23, 2.15) than their counterpart [Table 3]. Similarly, the odds of a chronic diabetes complication was higher among patient with a higher systolic BP ($\geq 140.00$ mm Hg.) or diastolic BP ($\geq 90.00$ mm Hg.) (AOR = 3.13; 95% CI 2.25, 4.35) and (AOR = 1.39; 95% CI 1.06, 1.81) than participants with lower systolic BP (<139.99 mmHg.) or diastolic BP (<79.99 mm Hg.) respectively. Whereas, patients who were government employees (AOR = 0.48; 95% CI 0.26, 0.90), taking anti-platelet drugs (AOR = 0.29; 95% CI 0.16, 0.52) and anti-dyslipidemia drugs (AOR, = 0.54; 95% CI 0.35, 0.84) were less likely to develop the complication than the patients who were unemployed and did not take anti-platelet and anti-dyslipidemia, respectively [Table 3].

## Discussion

This study aimed to assess the magnitude of chronic diabetes complications and associated factors among diabetic patients attending the general hospitals in the Tigray region, Northern Ethiopia. The overall magnitude of chronic diabetes complications in this study was 54.0% (95% CI: 51.35, 57.10). This is consistent with studies from Bahir Dar, Northwest Ethiopia (54%), and China (52%) [12,21]. The finding of this study was lower than the magnitude reported in Libya(69%), Indonesia (69%), Gurage zone, South Ethiopia(61%), Saudi Arabia (66%) and Nepal (72%) [47–51]. The probable cause for the higher prevalence reported in other studies compared to our study could be due to the presence of age, diabetes duration, and BMI differences. For instance, a study in Indonesia reported that 46% of study participants were aged >60 years old; in Indonesia, Ethiopia, Saudi Arabia, and Nepal, patients with $\geq 5$ years of diabetes duration were 73%, 53%, 46%,46% and 91%, respectively; and participants from a study in Saudi Arabia and Nepal with normal BMI were 17% and 20% respectively.

However, our study showed that 34%, 45% and 77% of the study participants were >60 years of age, with $\geq 5$ years of diabetes duration and normal BMI respectively. However, the result of this study is higher than a study in Gurage Zone, Southwest Ethiopia (46%). [21] The probable reason for the higher prevalence reported in our study compared to a study in Gurage Zone, south Ethiopia, could be due to ample size, study setting, and population variation. The study was done using a very small sample size and conducted in both the primary and general hospitals among type 1 and type 2 patients.

In this study, 27.28% (95% CI: 24.71, 29.85) participants had diabetes-related hypertension which is in line with a study done in Jimma, Ethiopia(25%) [15]. This is higher than studies from Palestine (23%), the town of Hosaena, Southern Ethiopia (23.9%) and Saudi Arabia (20%) [52–54]. The reason for the high prevalence in our study as compared to studies conducted in Palestine, Ethiopia and Saudi Arabia might be because of differences in the male-to-female ratio, time at which the study was conducted and study area. The result of this study is lower than studies conducted in Bangladesh (82%), Iraq (38%), Libya (33%), West Ethiopia (42%) and Jimma, Ethiopia (42%) [15,47,55–57]. The variation might have occurred because

**Table 3.** Factors associated with chronic diabetes complication among diabetic patients attending the general hospitals in Tigray region, Northern Ethiopian, 2019/2020 (N = 1,158).

| Variable | Category | Chronic DM complication | | OR (95% CI) | | |
|---|---|---|---|---|---|---|
| | | No | Yes | COR | P | AOR |
| Age | ≤40 year | 88 (7.6%) | 49(4.2%) | 1 | | 1 |
| | 41–45 year | 59(5.1%) | 41(3.5%) | 1.2(0.73, 2.12) | 0.423 | 1.26(0.70, 2.26) |
| | 46–50 year | 96(8.3%) | 89(7.7%) | 1.6(1.05, 2.61) | 0.163 | 1.44(0.86, 2.41) |
| | 51–55 year | 76(6.6%) | 76(6.6%) | 1.7(1.12, 2.88) | 0.399 | 1.26(0.73, 2.20) |
| | 56–60 year | 93(8.0%) | 103(8.9%) | **1.9(1.27, 3.11)** | 0.208 | 1.39(0.83, 2.35) |
| | ≥61 year | 118(10.2%) | 270(23.3%) | **4.1(2.72, 6.19)** | **0.000** | **2.51(1.50, 4.18)**\*\*\* |
| Marital status | 1. Single | 39(3.4%) | 38(3.3%) | 1.0(0.66, 1.70) | 0.221 | 1.51(0.77, 2.94) |
| | 2. Married | 380(32.8%) | 394(34.0%) | **1.8(1.03, 3.25)** | 0.926 | 1.01(0.67, 1.53) |
| | 3. Divorced | 47(4.1%) | 84(7.3%) | **1.7(1.04, 3.08)** | 0.059 | 1.68(0.98, 2.89) |
| | 4. Widowed | 64(5.5%) | 112(9.7%) | 1 | | 1 |
| Educational status | 1. Illiterate | 246(21.2%) | 339(29.3%) | **1.4(0.98, 2.02)** | 0.142 | 0.64(0.36, 1.15) |
| | 2. Primary school | 118(10.2%) | 134(11.6%) | 1.1(0.77, 1.74) | 0.210 | 0.69(0.39, 1.22) |
| | 3. secondary school | 89(7.7%) | 80(6.9%) | 0.9(0.59, 1.43) | 0.238 | 0.71(0.40, 1.24) |
| | 4. College/University | 77(6.6%) | 75(6.5%) | 1 | | 1 |
| Occupation | 1. Farmer | 119(10.3%) | 129(11.1%) | **0.6(0.47, 0.92)** | 0.946 | 0.98(0.62, 1.54) |
| | 2. Gov't employee | 103(8.9%) | 62(5.4%) | **0.3(0.25, 0.54)** | **0.023** | **0.48(0.26, 0.90)**\* |
| | 3. Private work | 135(11.7%) | 146(12.6%) | **0.6(0.48, 0.91)** | 0.552 | 0.87(0.57, 1.35) |
| | 4. Retired | 42(3.6%) | 78(6.7%) | 1.1(0.74, 1.76) | 0.532 | 1.19(0.68, 2.09) |
| | 5. Unemployed | 131(11.3%) | 213(18.4%) | 1 | | 1 |
| Monthly income (UD) | < $34.23 | 157(13.6%) | 234(20.2%) | **1.6(1.07, 2.68)** | 0.340 | 1.35(0.72, 2.50) |
| | 34.26–171.16 | 324(28.0%) | 351(30.3%) | 1.2(0.79, 1.91) | 0.356 | 1.28(0.75, 2.19) |
| | >171.16 | 49(4.2%) | 43(3.7%) | 1 | | 1 |
| BMI | < 25 kg/m2 | 466(40.2%) | 420(36.3%) | 1 | | 1 |
| | ≥ 25 kg/m2 | 162(14.0%) | 110(9.5%) | **1.3(1.00, 1.74)** | 0.994 | 0.99(0.72, 1.38) |
| Diabetes treatment regimen | Inulin (injectable) | 78(6.7%) | 61(5.3%) | **0.6(0.46, 0.94)** | 0.754 | 0.93(0.60, 1.44) |
| | Inulin & OHA\* | 33(2.8%) | 72(6.2%) | **1.8(1.19, 2.84)** | **0.000** | **2.45(1.49, 4.01)**\*\*\* |
| | OHGA\* | 419(36.2%) | 495(42.7%) | 1 | | 1 |
| Anti-platelet drug | Ye | 20(1.7%) | 116(10.0% | **1** | | 1 |
| | No | 510(44.0%) | 512(44.2%) | **0.1(0.10, 0.28)** | **0.000** | **0.29(0.16, 0.52)**\*\*\* |
| Anti-dyslipidemia drug | Ye | 45(3.9%) | 150(13.0%) | 1 | | 1 |
| | No | 485(41.9%) | 478(41.3% | **0.2(0.20, 0.42)** | **0.006** | **0.54(0.35, 0.84)**\*\* |
| Duration of diabetes | < 5 year | 345(29.8%) | 287(24.8%) | 1 | | 1 |
| | ≥ 5 year | 185(16.0%) | 341(29.4%) | **2.2(1.74, 2.81)** | **0.001** | **1.56(1.18, 2.05)**\*\* |
| Systolic blood pressure (SBP) | < 139.99 mmHg | 459(39.6%) | 400(34.5%) | 1 | | 1 |
| | ≥140.00 mmHg | 71(6.1%) | 228(19.7%) | **3.6(2.73, 4.96)** | **0.000** | **3.13(2.25, 4.35)**\*\*\* |
| Diastolic blood pressure (DBP) | < 89.99 mmHg | 503(43.4%) | 551(47.6%) | 1 | | 1 |
| | ≥ 90.00 mmHg | 27(2.3%) | 77(6.6%) | **1.6(1.27, 2.03)** | **0.016** | **1.39(1.06, 1.81)**\* |
| Pill burden | <4 pills/day | 360(31.1%) | 304(26.3%) | 1 | | 1 |
| | ≥ 4 pills/day | 170(14.7%) | 324(28.0%) | **2.2(1.77, 2.87)** | **0.001** | **1.63(1.23, 2.15)**\*\* |
| High saturated fat consumption | < 20 gm. fat/day | 143(12.3%) | 208(18%) | 1 | | 1 |
| | ≥ 20 gm. fat/day | 387(33.4%) | 420(36.3%) | **0.7(0.57, 0.96)** | 0.111 | 0.78(0.58, 1.05) |
| Physical activity | Inactive | 174(15.0%) | 274(23.7%) | **1.5(1.24, 2.01)** | 0.070 | 1.31(0.97, 1.77) |
| | Active | 356(30.7%) | 354(30.6%) | **1** | | 1 |

\*OHGA: oral hypoglycemic agent

\*significant at p<0.05

\*\*significant at p<0.01

\*\*\*significant at p< 0.0001., Homer-Leme how goodness-of-fit (P = 0.0063), BMI: Body Ma Index.

in this study only participants who became hypertensive after the occurrence of diabetes were considered, whereas the other studies may have included all participants who developed hypertension before and after the diagnosis of diabetes. Moreover, the presence of age, diabetes duration, and BMI differences may contribute to such variation.

In this study, peripheral vascular disease was seen among 9.13% (95% CI: 7.49, 10.81) participants, which is higher than the finding from a survey conducted in Sri Lanka (4.7%) [32,58]. The reason for the higher value of our study might be because of genetic variability, patients' poor adherence to medication, and practice related to lifestyle recommendations. However, the result of this study is lower than India (12%), Bangladesh (14%) and Libya (15%) [59–61]. This variation occurred due to our study population remained without a diagnosis for years because of a lack of awareness and access to advanced diagnostic tests.

Coronary artery disease (CAD)occurred among 3.28% (95% CI 2.25, 4.30) participants in this study, which is lower than studies done in Sri Lanka (11%), Saudi Arabia (23%), Bangladesh (26%), India (8%), Nepal (23%), and Iraq (15%) [32,60,62–64]. The reasons for the lower result of this study compared to the other studies are our study participants might have had a late diagnosis of diabetes, late initiation of treatment, or lifestyle modification, and a lack of access to advanced diagnostic studies like stress tests, echocardiography and nuclear perfusion imaging test. Moreover, genetic and racial variability may play a role in such discrepancy.

Of the total participants in this study, 4.31% (95% CI 3.14, 5.49) had a stroke, which is higher than studies done in Libya (1.9%), Saudi Arabia (0.19%), Nepal (1%) and Iraq (0.7%) [47,51,54,56]. The reason for the higher value of our result might be due to over-reporting, a larger sample size, and the late initiation of diabetes treatment or lifestyle recommendations compared to other studies, but this result is lower than India (7%), Bangladesh (11%), and Indonesia (18%) [59,60,65]. The reason for our study's lower figure could be participants in other studies were older, had a higher rate of obesity, and had diabetes for a longer time. In addition, socio-cultural variations and genetic predisposition play a role in the variation.

Peripheral neuropathy was seen among 11% (95% CI 8.84, 12.39) participants; this finding is almost similar to that of studies conducted in India (11%) and West Ethiopia (10%) [66,67]. The reason for the similarity could be due to the population characteristics, mainly the socio-demographic characteristics, being relatively similar.

This result is higher than the result of Saudi Arabia (1.4%) and Iraq (6.5%) [54,56]. The reason for the higher figure in this study could be due to socio-demographic characteristics variation; for instance, in this study, only T2D patients were included, and 28% of participants were from the rural area, whereas the study from Iraq included both types 1 and type 2 diabetes, and more than 53% of participants in the Saudi Arabia study were rural resident.

However, it is lower than the result of Sri Lanka (63%), Southwest Ethiopia (15%), Nepal (15%), Bangladesh (28%), India (19%), Tanzania (29%), and Egypt (22%) [32,49,51,60,62,68,69]. The probable reason for the lower figure in our study could be the lower rate of progression and the low onset of the disease, which discourage patients from seeking treatment early. In addition, a lack of routine foot examinations by senior experts could result in underreporting or miss diagnosis of such cases.

In this study the magnitude of diabetes nephropathy was 19.17%(95%CI: 16.90,21.44) this figure is in line with the study conducted in Ethiopia [15]. Unlike our study, Sri Lanka (51%), Nepal (25%), Bangladesh (43%), India (41.1%), Egypt (67%) and Sudan (39%) Studies have found higher percentages of participants with nephropathy [32,51,60,62,69,70]. The reason for the lower finding of our study might be ethnicity, racial, and socio-demographic characteristics difference, which play a major role in the development of diabetic nephropathy. However, our result is higher than the findings of studies in Tanzania (12%), China (11%), India (10.5%), Ethiopia (11.4%), Saudi Arabia (4.2%), and Iraq (14.2%) [12,21,63,64,66,68]. The

higher result in our study could be because in this study all forms of kidney disease were included, whereas in the other studies, microalbuminuria, macro albuminuria and high level of serum creatinine were reported separately. Moreover, genetic and socio-demographic variability may play a role.

In this study, retinopathy was seen in 9% (95% CI: 7.25–10.53) of participants, which is in line with studies conducted in Indonesia (7%), Tunisia (8.1%), Iraq (7.5%), and Ethiopia (10%) [16,64,65,69] and higher than a survey done in India (4.8%) and Saudi Arabia (3%) [63,66]. The probable reason for the higher value in this study could be attributed to the lack of awareness among our patients about the importance of regular eye examinations. Moreover, the patient's poor medication adherence, knowledge, and attitude might have contributed to such a difference.

However this result is lower than research findings done in Ethiopia (26%), Sri Lanka (26%), SSA (15%), Nepal (29%), Korea (38%), India (15.4%), Bangladesh (38%), Libya (31%), Tanzania (50%), Sudan (14%), and Egypt (21%) [21,32,35,51,58–61,68,70,71]. The rationale for the lower figure of this study compared to the other studies might be the lack of access to the regular dilated fundus examination technique due to it is not available at general hospitals. Hence, diabetic patients rarely check on their visual status in the absence of a symptom.

In this study, 4.14% (95% CI 2.99, 5.29) of the participants had a diabetes-related foot ulcer, which is similar to the study finding in Sri Lanka (2.6%), Korea (4.4%), Saudi Arabia (2.17%), Nepal (5%), and Ethiopia (5%) [15,32,51,58,63] and higher than China, Iraq (0.8%), Libya (1.1%), Sri-Lanka (1.3%), and Ethiopia. [12,32,61,64,67] (1.2%). This difference may be due to the small sample size and study population variation. However, this finding is lower than studies done in the North, South, and BLH, Ethiopia (21.2%, 20.4%, and 17%, respectively) [16,21,49]. The justification for the lower finding of our study could be due to underreporting and patients were not received regular foot examinations by a senior expert.

In this study, age, occupation, diabetes treatment regimen, anti-platelet drug, anti-dyslipidemia drug, duration of diabetes, systolic BP, diastolic BP and Pill burden were variables showing statistical association with the occurrence of diabetes complications. Similarly, the age of the participant, diabetes treatment regimen and/or duration of diabetes were also identified as factors associated with chronic diabetes complications in studies in, China, Ethiopia, and Libya [12,21,61].

However, gender, marital status, BMI, poor glycemic control, dyslipidemia, and family history of diabetes do not show association in this study, but gender in Libya and Iraq [61,64], marital status and BMI in Ethiopia [49], poor glycemic control in Ethiopia and Libya [49,61], dyslipidemia [61] and family history of diabetes in Libya [61] were identified as factors associated with chronic diabetes complication.

Some of the major limitations of this study that should be mentioned are: First of all, as it was hospital-based cross-sectional in its design, this only allows for the identification of variables that have an association with the dependent variable rather than causation. Second, it could not be generalizable to the entire population and to diabetic patients who don't receive care in the study area's public general hospital. Third, some independent variables, such as lipid profile, HgbA1C, and eating pattern for each day, were not taken into consideration. Finally, the target population of this study included diabetic patients who were treated at the general hospital by physicians with limited competence to diagnose and the lack of some diagnostic tests even though the patient had a significantly more complex disease burden.

## Conclusion

In this study, more than half participants had at least one chronic diabetes complication. Diabetes-related hypertension was the most prevalent type of chronic diabetes complication,

followed by eye and renal disease. Furthermore, chronic diabetes complication was associated with age, occupation, diabetes treatment regimen, an anti-platelet drug, anti-dyslipidemia drug, diabetes duration, systolic BP, diastolic BP, and pill burden. Therefore, healthcare providers managing patients with diabetes should work collaboratively with other stakeholders to ensure that all patients with diabetes are screened for early detection and treatment of complications of diabetes using different approaches and strategies. A future follow-up study should be carried out to investigate the cause of chronic diabetes complications among diabetic patients.

## Supporting information

**S1 Data set.**
(SAV)

## Acknowledgments

We would like to acknowledge all staff of referral clinic of the elected public general hospital for their support, the participant for kindly giving the required information, supervisor and data collector.

## Author Contributions

**Conceptualization:** Kalayou K. Berhe.

**Formal analysis:** Kalayou K. Berhe.

**Methodology:** Kalayou K. Berhe, Lilian T. Mselle, Haftu B. Gebru.

**Supervision:** Lilian T. Mselle, Haftu B. Gebru.

**Writing – original draft:** Kalayou K. Berhe.

**Writing – review & editing:** Lilian T. Mselle, Haftu B. Gebru.

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
