## [Decision Letter · Decision Letter 0]

9 Mar 2023

PONE-D-22-12568Magnitude of  Chronic diabetes complication and its associated factors among adults with type 2 diabetes in Tigray region, northern EthiopiaPLOS ONE

Dear Dr. Berhe,

Thank you for submitting your manuscript to PLOS ONE. After careful consideration, we feel that it has merit but does not fully meet PLOS ONE’s publication criteria as it currently stands. Therefore, we invite you to submit a revised version of the manuscript that addresses the points raised during the review process.

We look forward to receiving your revised manuscript.

Kind regards,

Manal S. Fawzy, Ph.D., M.D.

Academic Editor

PLOS ONE

Journal Requirements:

5**. **Please include a separate caption for each figure in your manuscriptijol.

Additional Editor Comments (if provided):

Based on the reviewers' feedback and the editorial assessment, several concerns should be addressed by the authors.

Reviewers' comments:

Reviewer's Responses to Questions

**Comments to the Author**

1. Is the manuscript technically sound, and do the data support the conclusions?

Reviewer #1: Partly

Reviewer #2: Yes

2. Has the statistical analysis been performed appropriately and rigorously? 

Reviewer #1: Yes

Reviewer #2: Yes

3. Have the authors made all data underlying the findings in their manuscript fully available?

Reviewer #1: Yes

Reviewer #2: Yes

4. Is the manuscript presented in an intelligible fashion and written in standard English?

Reviewer #1: No

Reviewer #2: No

5. Review Comments to the Author

Reviewer #1: This article can be made simpler for the scientific community with some modifications and clarifications as below.

Minor comments

1. Consistently use either “Tigray” or “Tigrai” in the title, author affiliation, study area, and elsewhere.

2. Change “north Ethiopia / northern Ethiopia” to “Northern Ethiopia” throughout the manuscript

3. Instead of saying "an institutional-based study," say “a multi-center cross-sectional study." since the study was conducted in ten public general hospitals.

4. It is better to say 54% rather than 54.0%, and 27% rather than 27.0%.

5. Throughout the manuscript, apply comma for numbers with more than two digits, like instead of saying 60 000–100 000 people, better to say 60,000-100,000 people.

6. Remove double full stops throughout the manuscript

7. Since it is an international journal, better to change the income status that you mentioned in Ethiopian birr (ETB) to US dollar using the exchange rate at the date of data collection.

8. I recommend you to avoid writing formulas for sample size calculations, it would be better to make it in sentence format the approach you applied for calculating the sample size than writing detailed mathematical equations.

9. Remove the Hosmer-Lemeshow goodness-of-fit test result you mentioned in the method section and put it at the result section (particularly at the multivariable regression table), At the method section you have to mention what you have did not what you found.

10. Add your response rate in the result section, as your study is a cross-sectional study

11. I appreciate writing of the findings of the measure of effect (AOR with its 95% CI), make it consistent in the result section of the manuscript (you mentioned [AOR (95% CI) = 0.48(0.26-0.90), P=0.023]), re-write it as you have rightly described in the abstract section.

12. Avoid P-value if used 95% CI instead of writing like ([AOR (95% CI) = 0.48(0.26-0.90), P=0.023]), remove the P-value and re-write it as (AOR=0.48; 95% CI 0.26,0.9), and remove the hyphen (-), rather use comma for writing the 95% CI.

13. Check the reference section, the majority of the citations are appropriate, but there are some references citrated in appropriately, check them again using software’s or manually

14. Make sure whether your questionnaire has three or four parts, there are inconsistencies in their parts mentioned in the data collection tool and measurement part.

Major comments

1. Rationale of the study: You mentioned that the previous studies had a critical methodological limitation, which hindered the scientific community to make any conclusion or judgment? How do you know whether the sample size was small, if appropriately calculated even 10 sample size can be enough? There are more than ten similar articles published elsewhere in the country, including in the region, what was the added value of your study? I would suggest you to re-write it again, considering convincible scientific argument.

2. Did you assess the risk factors? What you assessed was factors associated with chronic diabetic complications, do you think that we can interchangeably use risk factors and factors associated? Can we assess risk factors using simple, classical cross-sectional study design? I recommend you to consistently use the term factors associated, not risk factors in your manuscript.

3. Sample size calculation: It is appreciable that you have used 3% margin of error to maximize the sample size, and your response rate was 94.6%, but after adding 10% non-response rate, the final sample size should be 1,178 not 1,061.

4. Rewrite incomplete sentences, as there are many incomplete sentences, check the spelling and grammar issues, please check again the whole manuscript for spelling and grammar issues (use of present or past tense), I am not comfortable with the write-up.

Reviewer #2: PLOS ONE

PONE-D-22-12568

Research Article

Magnitude of Chronic diabetes complication and its associated factors among adults

with type 2 diabetes in Tigray region, northern Ethiopia

By: kalayou kidanu Berhe,

Mekelle University College of Health Sciences

Mekelle, Tigray ETHIOPIA

Dr Hussein Ismail, Reviewer report to PLOS one December 2022

1. There is a lot of English language errors that need to be corrected, I believe the manuscript needs a professional proofreading before publications. A lot of errors are identified as in the title Chronic (is written with capital letter and it should not).

2. Moreover, there are a lot of abbreviation errors, that need to be corrected, e.g., in the abstract Bsc nurses, OHA, …etc. The manuscript has a lot of abbreviation errors as well, e.g., IDF in the abstract.

3. Title: Magnitude of Chronic diabetes complication and its associated factors among adults

with type 2 diabetes in Tigray region, northern Ethiopia.

The authors stated that they studied 10 general hospitals out of 13 hospitals, and they did not included any referral or primary care centers. So, the selection is based on general hospitals only, there fore it should be mentioned in the title.

My suggestion for the title:

Magnitude of chronic diabetes complication and its associated factors among diabetic patients attending

the general hospitals in Tigray region, northern Ethiopia.

4. METHODS

4.1. Why did not the author include all the 13 general hospitals? I was surprised of taking 10 hospitals and leaving 3 hospitals. Please explain.

4.2. Sampling:

The methods of sampling were explained efficiently. Although, the author in included p=0.535, (p=proportion of chronic diabetes complications) please include the refence you used that stated the complications proportion as 0.535. Sampling is a step the author did before the research; I am surprised that this proportion used in the sampling was 0.535 is the same as the magnitude of diabetes complications which was the main finding of this study. Please explain.

4.3. Regarding the operational definitions, the definition of hypertension the author used was BP> 140/90, I checked the reference used and it was outdated. Reference number 30.

30. Muxfeldt ES NAdR, Salles GF, Bloch KV. Demographic and clinical characteristics of hypertensive patients in the internal medicine outpatient clinic of a university hospital in Rio de Janeiro. Sao Paulo Med J Child Adolesc Behav. 2004;122:87-93.

My suggestion: please update all the operational definitions according to the updated guidelines or manuscript. Regarding hypertension, you may use the American heart association guidelines 2017 or the European Society of cardiology (ESC) guideline 2018. I recommend the ESC because it agrees with the level of 140/90 that you chose.

Moreover, according to guidelines: No caffeine, no smoking, no eating for at least 2 h before measurement. The author stated BP was measured after30 minutes after hot drink as coffee. Please, explain and what is the refence you used?

5. RESULTS

5.1. The author stated in the results

• (….. in which 29.4%, 34.0%, 47.1% and 52.4% participants had chroic diabets complication respectively.)

• in which 42.7%, 10.0% and 13.0% had chronic diabetes complication

respectively.

• had DBP of > 90.00 mmHg in which 6.0%, 32.2% , 19.7% and 6.6%

participants had at least one chronic diabetes complication respectively.

Suggestion:

• Please specify each complication associated with these numbers.

• Please apply this notion along the whole paper.

5.2. Tables: The authors need to put all the abbreviations in the footnote related to each table. Some abbreviations are missing in the footnotes.

6. DISCUSSION

The discussion is well written.

7. CONCLUSION

It highlights the main findings and supported by the study results.

8. Reference

The authors included a lot of outdated refences. As the refences included:

• Reference 10: 1996

• Reference 22: 1965

Suggestion: updating the refences accordingly.

Regarding recent refences: Only one reference (number 35) was published 2020.

6. PLOS authors have the option to publish the peer review history of their article (what does this mean?). If published, this will include your full peer review and any attached files.

Reviewer #1: **Yes: **Zenawi Hagos Gufue, Adigrat University, Ethiopia

Reviewer #2: **Yes: **Hussein M. Ismail, MD Cardiology

---

## [Author Response · Author response to Decision Letter 0]

11 Apr 2023

RESPONSE TO REVIEWERS

Response to Reviewer #1

I. Minor comments 

1. “Tigray” is used consistently instead of “Tigrai” in the title, author affiliation, study area, and elsewhere in the manuscript ……………………………………………………………………………………………….(Page all)

2. “north Ethiopia / northern Ethiopia” was changed to “ Northern Ethiopia”…………………………(Page all)

3. "an institutional-based study," was replaced by “a multi-center cross-sectional study”…………….(Page1,4)

4. It was corrected as 54% from 54.0% and as 27% from 27.0% and other similar issues corrected throughout the document as per your suggestion .................................................................................................(Page: 2, 8,9,11)

5. Comma was used for numbers with more than two digits (e.g. 60000–100 000 was edited as 60,000-100,000)………...(page all) 

6. Double full stops were removed………………………………………………………………………...(page all)

7. The mentioned Ethiopian birr (ETB) in the Monthly income status section(Tables 1,3) was changed in to US dollar based on average exchange rate of 2019…………………………………………………………(Page 9)

8. The formula for sample size calculation was removed and rewrote in sentence format as per your recommendation and checked using StatCalc for population survey via Epi Info 7.0 software………..(Page4,5)

9. The result of Hosmer-Lemeshow goodness-of-fit test was removed from the method section and placed at the result section (Factors associated with chronic diabetes complication & table3 foot note), ……….(Page11)

10. Result of response rate is included/added in the result section of the manuscript………………….........(Page 8)

11. The multi variable analysis result was re-wrote as per your recommendation……………………...(Page 11, 12)

12. In Abstract & result section P-value was removed and re-wrote as (AOR=0.48; 95% CI 0.26, 0.9), and the hyphen (-) was removed instead comma was use for writing the 95% CI……..............................(Page 2, 11,12)

13. In appropriately cited references were checked & corrected via endnote software ...………………(Page 18-21)

14. The questionnaire has four parts, then corrected as “ it has four parts”…………………….……….....(Page 5)

II. Major comments

1. Rationale of the study: revised and additional scientific arguments are included ………………(Page 2,3,4)

2. The term “Risk factor” was replaced by “factors associated” and used consistently throughout the manuscript ………………………………………………………………………………………………(Page all)

3. Sample size calculation: …………………………………………………………………………...(Page 2.5,8)

The required sample size (n) was estimated manually using a single population proportion formula and cheeked using STATCALC for population survey via Epi-info version 7 software with assumptions of 95% CI (z = 1:96), d=0.03, and P=0.535.

Therefore the initial sample size was 1061.882 ( ni = (Z1-α/2)2p (1–p)/d2= (1.96)20.535(1-0.535)/(0.03)2 ). However, a refusal rate of 10% (1061.882*0.1) =106.1882 was added and gives a final sample size of 1,168.0702 (1,061.882+106.1882), 

Accordingly10 questioners were excluded because of gross incompleteness and 1,158 participants’ questioner were fit for final analysis which makes response rate of 99.14 % ……………………………… 

4. Incomplete sentences were re-wrote, spelling and grammar issues were checked and corrected…(Page all) 

Response to Reviewer #2: 

Title & abstract 

1. Gross English language Proofreading was done to correct errors of Spelling, grammar, punctuation and statement construction throughout the manuscript…………………………………………………. (Page all)

2. To avoid confusion with other similar abbreviations the expanded form of the abbreviations were included e.g. IDF (International Diabetes Federation ), Bsc is corrected as BSc and other abbreviations errors corrected accordingly ……………………………………………………………………………………..(Page 2,10,11 ) 

3. Research Title was modified as “Magnitude of chronic diabetes complication and its associated factors among diabetic patients attending the general hospitals in Tigray region, northern Ethiopia” based on your suggestion …………………………………………………………………………………………………………(page-1)

4. Method and materials 

4.1 Study Area:: the study was done at 10 general hospitals out of all 14 (not 13, it was written by mistake) general hospitals, all general hospitals were not included because of the four hospitals were exclude randomly due to budget constraint / logistic issue…………………………………………………...(Page4) 

4.2 Sampling: A reference (ref.no 21) for P=0.535, (p=proportion of chronic diabetes complications) was included and this figure is used to calculate the sample size which was done in 2015 which was before our study conducted (2019/20) and taken from a study done at FelegeHiwot referral hospital, Bahardar, Amhara region, Northwest Ethiopia…………………………………………………………………..(Page5)

As you mention, by chance the chronic diabetes complication proportion of the study done at FelegeHiwot referral hospital (P=0.535) which we use for sample size calculation is similar to our finding (P=0.54). This could occur because of similarity in socio-demographic characteristics, poor glycemic control as a result of poor adherence to diabetes self-management recommendations……………………… (Page5, 11)

Therefore, the finding of this study (overall magnitude of chronic diabetes complication was p=0.54 or 54% of the patients had at least one type of the complication…………………………………… (Page 11)

But both studies have difference in many things such as sample size (344 vs. 1,158), study facility (one tertiary hospital vs. ten general hospitals), study area (Amhara region vs. Tigray region) and study period (2015 Vs. 2019/20)……………………………………………………………………………….... (Page4-6)

4.3 Operational definition: 

4.3.1 Definition of Hypertension: The reference was updated and replaced by the reference that you recommend “European Society of cardiology (ESC) guideline 2018” (ref.no 33, 34)…………. (Page7,8)

4.3.2 Definition of other chronic diabetes complications: all definition/ diagnostic criteria of other chronic diabetes complications (operational definitions) were updated according to the updated guidelines or manuscript based on your suggestion……………………………………….(Page 7,8)

4.3.3 Data collection and measurement(BP): the timing of BP measurement related to coffee consumption was revised as 1-2 hours (Study indicated that after caffeinated beverage intake blood pressure changes occur within 30 minutes, peak in 1-2 hours, and may persist for more than 4 hours) and reference also included ………………………………………................................................................... (Page 6) 

5. Result 

5.1 Socio-demographic, Clinical and behavioral characteristics: 

In this section we try to explain the findings based on the cross-tabulation analysis results but as mentioned in your review 

• In which 29.4%, 34.0%, 47.1% and 52.4% participants had chronic diabetes complication respectively”

• In which 42.7%, 10.0% and 13.0% had chronic diabetes complication respectively.

• In which 6.0%, 32.2%, 19.7% and 6.6% participants had at least one chronic diabetes complication respectively.

Those are findings of overall chronic diabetes complication in terms of socio-demographic, clinical and behavioral characteristics in the cross-tabulation analysis result (Table 1,2) but we observed such way of explanation may result in confusion for the reader, so to make it clear and simple we prefer to omit/ remove all cross-tabulation findings of chronic diabetes complication from the text explanation of socio-demographic, clinical and behavioral characteristics section but readers can get those findings from the Table 1 & 2 …………………………………………………………………………………...(Page8-11)

5.2 Tables: all the abbreviations related to each table were placed in the footnote…… (Page 9,10,11,13)

6. Discussion: Except English language Proofreading, revision was not done in this section because you mentioned as “The discussion is well written”

7. Conclusion: Revision was not done because you stated as “It highlights the main findings and supported by the study results”

8. Reference: Reference 10: was replaced with updated reference, Reference 22: was replaced with updated reference. Accordingly all references were updated as per your suggestion ………(Page 18-21)

---

## [Decision Letter · Decision Letter 1]

16 May 2023

PONE-D-22-12568R1Magnitude of chronic diabetes complications and its associated factors among diabetic patients attending the general hospitals in Tigray region, Northern EthiopiaPLOS ONE

Dear Dr. Berhe,

Thank you for submitting your manuscript to PLOS ONE. After careful consideration, we feel that it has merit but does not fully meet PLOS ONE’s publication criteria as it currently stands. Therefore, we invite you to submit a revised version of the manuscript that addresses the points raised during the review process.

We look forward to receiving your revised manuscript.

Kind regards,

Manal S. Fawzy, Ph.D., M.D.

Academic Editor

PLOS ONE

Journal Requirements:

Reviewers' comments:

Reviewer's Responses to Questions

**Comments to the Author**

1. If the authors have adequately addressed your comments raised in a previous round of review and you feel that this manuscript is now acceptable for publication, you may indicate that here to bypass the “Comments to the Author” section, enter your conflict of interest statement in the “Confidential to Editor” section, and submit your "Accept" recommendation.

Reviewer #2: All comments have been addressed

Reviewer #3: All comments have been addressed

2. Is the manuscript technically sound, and do the data support the conclusions?

Reviewer #2: Yes

Reviewer #3: Yes

3. Has the statistical analysis been performed appropriately and rigorously? 

Reviewer #2: Yes

Reviewer #3: Yes

4. Have the authors made all data underlying the findings in their manuscript fully available?

Reviewer #2: Yes

Reviewer #3: Yes

5. Is the manuscript presented in an intelligible fashion and written in standard English?

Reviewer #2: No

Reviewer #3: Yes

6. Review Comments to the Author

Reviewer #2: First, the manuscript looks much better than the first version, thanks for the authors.

Although, I do not think it is ready for publication.

The most important is to revise the enligh language again, and the author should submit an offcial proofreading certifcate for the manuscript, if the Editorial Board advise on this regard, it will be very helpful. As I still see lot of English and Grammer mistakes, and the writing way is not quite professional.

1. The abstarct/conclusion:

Conclusion: In this study, the magnitude of chronic diabetes complication was higher because more than half

of the study participants had at least one complication.

I donot understand this statement, the magitude is high than...what?

Also, revise the conculsion

2. The exclusion and inclusion criteria:

The author put bothe the criteria togther, which is confusing. Please specifiy what are the inclusion criteria? and what are the eclusion criteria?

3. Diagnosis of chronic diabetes complication:

3.1. I suggest using this statment in stead of yours

Coronary artery disease (CAD): The diagnosis criteria for CAD were either a patient with typical anginal pai or equivalent

symptoms and an abnormal resting ECG or an asymptomatic patient with abnormal stress test, either by

ECG or echo or a nuclear perfusion imaging test .

3.2. Peripheral vascuar disease:

The peripheral vascular disease defintion, it is advised to limit the ABI to less than 0.9 only, and delete more than 1.3

3.3. Neuropathy

loss of sensitivity is mis nomer, replace with hyposthesia or anasthesia in lower and upper limbs

spelling: limp - limb.

4. Tables:

4.1. Yes column: yes is wrongly written. Plz, correct.

4.2: The abbreveiations shoud be consistent: if you use SBP for systolic blood pressure, you have to use DBP for diastolic blood pressue, plz be consistent along the whole manuscript

Reviewer #3: I thank the authors to conduct this interesting study at the local setting.

All comments have addressed. No further comment required.

7. PLOS authors have the option to publish the peer review history of their article (what does this mean?). If published, this will include your full peer review and any attached files.

Reviewer #2: **Yes: **Hussein M Ismail

Reviewer #3: **Yes: **Mohammed Abdu Seid

---

## [Author Response · Author response to Decision Letter 1]

17 Jun 2023

RESPONSE TO REVIEWERS

Response to Reviewer #2

1. Abstract 

1.1 Conclusion: revision was made based on your comment and it is revised as “In this study, more than half participants had at least one chronic diabetes complication”…………………………………....(Page 2)

2. Method 

2.1 The exclusion and inclusion criteria: the inclusion criteria and exclusion criteria are separately written under eligibility criteria to avoid confusion………………………………………………………...(Page5) 

2.2 Diagnosis of chronic diabetes complication

2.2.1 Coronary artery disease (CAD): corrected as per your suggestion……………………….......(Page 7

2.2.2 Peripheral Vascular disease: as per your suggestion the more than 1.3 (>1.3) was deleted from the definition, ABI to less than 0.9 only is used ………………………………………………………..(Page7)

2.2.3 Neuropathy: misnomer of loss of sensitivity in the definition replaced with “ hyposthesia or anesthesia” in lower and upper limbs & the work limp is corrected as limb……………………...(Page7)

3. Results 

3.1 The wrongly written the word yes in the column is corrected throughout the tables………(Page 9,10,12 )

3.2 All abbreviations (SBP, DBP) were written consistently throughout the document ………...(Page 10-17) 

4. Conclusion: corrected as “In this study, more than half participants had at least one chronic diabetes complication”…………………………………………………………………………………….........(Page15)

5. Language editing: Our Manuscript was copyedit for language usage, spelling, and grammar by Zainabu Karim Mohamed from MUHAS, Tanzania (email: zainab.karim4@gmail.com) and Prof. Lilian T. Mselle (email: nakutz@yahoo.com ) from MUHAS, Tanzania. Uploaded as Supporting Information file 1 ……………………………………………………………………………………………………….(all page)

---

## [Editor Report · Decision Letter 2]

7 Aug 2023

Magnitude of chronic diabetes complications and its associated factors among diabetic patients attending the general hospitals in Tigray region, Northern Ethiopia

PONE-D-22-12568R2

Dear Dr. Berhe,

We’re pleased to inform you that your manuscript has been judged scientifically suitable for publication and will be formally accepted for publication once it meets all outstanding technical requirements.

Kind regards,

Manal S. Fawzy, Ph.D., M.D.

Academic Editor

PLOS ONE

Additional Editor Comments (optional):

The authors have adequately addressed the concerns raised by the reviewers. Thank you
---

## [Editor Report · Acceptance letter]

17 Aug 2023

PONE-D-22-12568R2 

The magnitude of chronic diabetes complications and its associated factors among diabetic patients attending the general hospitals in Tigray region, Northern Ethiopia 

Dear Dr. Berhe:

I'm pleased to inform you that your manuscript has been deemed suitable for publication in PLOS ONE. Congratulations! Your manuscript is now with our production department. 

Kind regards, 

on behalf of

Professor Manal S. Fawzy 

Academic Editor

PLOS ONE